# An Investigation of the Nature of Fear within ACL-Injured Subjects When Exposed to Provocative Videos: A Concurrent Qualitative and Quantitative Study

**DOI:** 10.3390/sports10110183

**Published:** 2022-11-18

**Authors:** Cameron Little, Andrew P. Lavender, Cobie Starcevich, Christopher Mesagno, Tim Mitchell, Rodney Whiteley, Hanieh Bakhshayesh, Darren Beales

**Affiliations:** 1Curtin enAble Institute and Curtin School of Allied Health, Curtin University, Perth, WA 6102, Australia; 2Institute of Health and Wellbeing, Federation University Australia, Ballarat, VIC 3350, Australia; 3Institute for Health & Sport, Victoria University, Melbourne, VIC 3011, Australia; 4Pain Options, Perth, WA 6151, Australia; 5Aspetar Sports Medicine Hospital, Doha 29222, Qatar; 6Curtin School of Electrical Engineering, Computing and Mathematical Sciences, Curtin University, Perth, WA 6102, Australia

**Keywords:** knee injury, anterior cruciate ligament (ACL), fear, videos

## Abstract

Fear is a factor contributing to poor return to sport after an anterior cruciate (ACL) injury, however the identification and assessment of fear is challenging. To improve understanding of fear, this study qualitatively and quantitatively assessed responses to videos depicting threat to knee stability in people who had experienced an ACL injury. ACL-injured participants who had above average fear on the Tampa Scale of Kinesiophobia and were at least 1-year post-injury/surgery were eligible. Participants were shown four videos depicting sequentially increasing threat to their knee stability (running, cut-and-pivot, feigned knee injury during cut-and-pivot, series of traumatic knee injuries). Qualitative interviews explored participants feeling related to viewing the videos. Participants quantitatively self-rated fear and distress in response to each video. Seventeen participants were included in this study (71% female, with an average time since last ACL injury of 5 ½ years). Five themes were identified: (1) Evoked physiological responses, (2) Deeper contextualisation of the meaning of an ACL injury influencing bodily confidence, (3) Recall of psychological difficulties, (4) Negative implications of a re-injury, and (5) Change to athletic identity. Quantitatively, direct proportionality was noticed between threat level and reported fear and distress. Specifically, participants reported increasing levels of fear and distress as the videos progressed in threat level, with the largest increase seen between a cut-and-pivot movement to a feigned injury during a cut and pivot. The results support the notion that in addition to being a physical injury, an ACL injury has more complex neurophysiological, psychological, and social characteristics which should be considered in management. Using video exposure in the clinic may assist identification of underlying psychological barriers to recovery following an ACL injury, facilitating person-centred care.

## 1. Introduction

The anterior cruciate ligament (ACL) provides proprioceptive feedback and mechanical stability within thef knee [1]. It is particularly important in supporting the knee during dynamic pivoting, acceleration, deceleration and change of direction movements, which are often performed in sports [1]. Injuries to the ACL are frequent [2], with almost 200,000 ACL reconstructions in Australia between July 2000 and June 2015 [3]. These injuries appear to be increasing [4], with many people not returning to their prior level of functioning. In a systematic review of 69 articles reporting on 7556 participants following a surgically managed ACL injury, only 55% of individuals returned to their competitive sport despite achieving successful surgical outcomes [5]. In another systematic review of 19 studies totalling 2175 participants after a surgically managed ACL injury, 37% did not return to any sport [6], with 65% citing psychological reasons, and 77% citing fear of re-injury as their main concern for failure to return to sport. This aligns with other research which identifies the significant role of psychological factors on return to sport, and recovery in general after an ACL injury [7,8,9].

The experience of an ACL injury can be traumatic and may lead to symptoms similar to post-traumatic stress disorder [10]. Behaviours such as avoidance, hesitation to attend therapy, guarding or protecting the knee during movement, and suboptimal performance may all be sequelae of underlying emotional and/or psychological distress [10,11,12]. Coupled with negative self-belief about knee function, this may increase an individual’s knee-related fear. Fear, which can be defined as “a basic, intense emotion aroused by the detection of imminent threat” [13]. Fear of re-injury is one of the most commonly reported reasons for not returning to sport in individuals who have suffered an ACL injury [7,8,14,15,16,17].

Fear may be heightened, or attenuated, through subconscious and conscious learning mechanisms [18]. Fear conditioning refers to the subconscious process wherein a previously non-fearful stimulus, such as playing sport, becomes associated with an unconditioned fearful stimulus, such as a traumatic injury event. The fear of the unconditioned injury, conditions the individual to also fear the previously non-fearful sport [18]. A conditioned fear response may also increase a person’s distress. Distress, defined as “a general term to describe a state of emotional suffering that interferes with the level of functioning” ([19], p. 877), may produce subconscious defensive or avoidance behaviours, such as a fight or flight response [20]. Distress and fear, whilst likely having some conceptual overlap [21], may differ in the context of an ACL injured individual. These individuals may be suffering emotionally and avoiding return to activities ([19], p. 877), but not necessarily fearful of an imminent threat [13]. Conscious thoughts and experiences can also contribute to fear [22].

Researchers who investigated lower back pain found that the conscious processing of a threat (to physical well-being) is complex and both physical and psychosocial factors can contribute to a heightening, or reduction, of fear [23,24,25]. In disorders such as chronic lower back pain, fear of movement has been assessed by exposing participants to potentially threatening stimuli, including pictures [26] and videos [27,28,29]. For example, participants with lower back pain exposed to four-second videos depicting potentially hazardous movements (e.g., shovelling soil with a bent back) demonstrated significantly greater brain activity, using Functional Magnetic Resonance Imaging, in regions of the brain that processes fear (e.g., amygdala and left anterior insula) compared to viewing harmless activities [29]. In a study of an ACL-injured cohort, nearly half (45.5%) identified cutting (i.e., planting the foot and changing direction by 90°) as the most feared movement [30]. Exposing people who have suffered an ACL injury to videos depicting potentially threatening stimuli may evoke a fear response that could offer clinical insight.

This study exposed people minimum 1-year after an ACL-injury/surgery to a series of videos depicting sequentially increasing threat to knee stability. The qualitative arm explored participants feelings related to viewing the videos, whilst the quantitative arm explored participants self-rated fear and distress levels in response to the videos. The overarching purpose of this study was to provide further insight into the nature of fear after an ACL injury. In addition, we aimed to provide some initial indication of the viability of using a video medium to assess fear in people who had experienced an ACL injury in the return to sport phase of their rehabilitation.

## 2. Materials and Methods

### 2.1. Study Design

A concurrent qualitative and quantitative design was selected, with both data types interpreted and merged during the Discussion phase [31,32]. Fear and distress responses were documented in participants during a series of videos depicting increasing threat to the knee, specifically: running, a cut-and-pivot action, a cut and pivot + a feigned knee injury, and a series of video clips of traumatic knee injuries. Qualitative data was gathered from individual participant experiences and feelings when viewing the series of threatening movements, to allow for a deeper understanding of the fear and distress associated with ACL injuries. The qualitative component of the study was aligned to the Consolidated Criteria for Reporting Qualitative Research (COREQ) guidelines, a 32 item checklist designed for explicit, comprehensive reporting of qualitative data [33]. Quantitative data was collected through self-reported ratings for fear and distress following each video. Data integration of the qualitative and quantitative components was conceived across the design and commented on in the Discussion [34]. Consumer engagement for feasibility, appropriateness and effectiveness of study [35] with three ACL-injured individuals (31-year-old male, 32-year-old female, 33-year-old male, not included in Table 1) occurred in the development of the study. This involvement included assistance in the generation of the research question and piloting of the videos depicting the sequentially threatening movements, with the aim to ensure the appropriateness and safety of the videos for this study. In addition, after 12 interviews these consumers were also given the opportunity to comment on the results. This feedback indicated that results were representative of the engagement group beliefs.

### 2.2. Participants

Participant inclusion criteria were: aged 18–50, and either at least 1-year post injury without surgery or 1-year post ACL reconstruction surgery [36]. Surgically or conservatively managed participants were combined given similar long-term patient reported outcomes of knee function, activity levels, and quality of life [37,38,39,40,41,42,43]. Eligible participants had to report ‘above average’ fear through scoring more than 17 on a modified Tampa Scale Kinesiophobia (TSK). This scale has previously been used to identify fear of movement in an ACL reconstructed cohort, wherein the median score (average fear) equalled 17 [17] with those above considered to have above average fear. Potential participants were excluded for any of the following: non-English-speaking background, multiple pain sites, or low back pain with radicular signs.

### 2.3. Sampling and Power

The sample size was determined as the point where the qualitative data saturation occurred. This was determined by iteratively reviewing participant information to ensure the appropriateness and adequacy of collected data [44,45], with the depth and breadth of results determined to be deemed sufficient by the interdisciplinary team [45,46]. The nature of an iterative approach means there will be constant evolution of new themes. Saturation was reached after 16 interviews; one more interview was completed to ensure no new themes were discovered [45,46].

### 2.4. Measures

Participant profiling data included age, sex, year of the ACL injury(s), and if they had an ACL reconstruction. A modified TSK to include knee specific questions rather than pain was collected from participants as an indication of fear of movement [17]. The TSK was rated from 0–51, with a higher score indicating higher levels of fear. The Anterior Cruciate Ligament-Return to Sport Index (ACL-RSI) was used as an indication of psychological readiness to return to sport [47]. This scale is rated out of 100 with lower scores indicating higher impairment.

Participants were asked how much they had been ‘bothered’ by their knee in the last week across four constructs: (1) fear of movement, (2) distress/anxiety related to knee movements, (3) lack of confidence and (4) pain (rated from 0–10 where 0 was no bother at all and 10 was extremely bothered). The return to previous level of function was scored from 0 to 3 with: no return to any exercise = 0, return to basic exercise (gym, etc.) but no sport = 1, return to sport at a lower level than previous = 2, or return to previous level of sport = 3.

Standardised instructions, covering testing procedure and a safety caveat were provided prior to watching the first video, this included clear instructions that participants could stop testing at any time if they felt uncomfortable. Following the first video (running), quantitative data was collected by asking participants to self-rate their level of fear from 0 (no fear at all) to 10 (extreme fear). ‘Distress’ then replaced ‘fear’ in the question to attain a distress score. An 11-point scale was chosen to allow better discrimination between the participant scores [48]. Participants were then asked, ‘How do you feel about this video?’ to collect qualitative information.

The same procedure for quantitative and qualitative was followed after each subsequent video (a cut-and-pivot action, a cut and pivot + a feigned knee injury, and a series of video clips of traumatic knee injuries). After the four-video protocol was completed, participants were asked an open-ended question, “Did any of these videos make you feel any similar emotions or fear to what you experienced during your own ACL injury?” (referring to both the initial injury and the rehabilitation journey) with, a follow up question, “can you explain to me why/why not?”. The interviewer then allowed participant to guide the discussion and prompted any points requiring clarification.

At the cessation of testing, an informal discussion was undertaken for a minimum of 5 min, this was to ensure participants had not had any ongoing emotional effects as a result of watching the videos. During this time, participants were asked ‘how they were feeling’ and whether there was any ‘psychological aggravation’ as a result of the testing protocol. Participants were subsequently contacted within 24 h to ensure no aggravation had occurred. No participants had ongoing emotional effects after the testing.

### 2.5. Procedure

Participants were recruited between October and December 2020 using a social media advertisement shared via Facebook. The social media advertisement contained a link to an online questionnaire, which participants completed to determine their eligibility for the study. Key information on the study design was provided to allow participants to give informed consent. General principles of participant sampling were used to ensure the quality of the sample, including making the sampling feasible, affordable, and specific to the research question whilst acknowledging and minimising biases [49]. Non probabilistic purposive sampling was used to identify the participants, which consisted of ACL-injured individuals who could articulate their emotions. This technique identified information rich cases for the most effective use of resources [50]. No financial reward was offered to take part in the study.

Videos were created to assess participants’ fear and distress by exposing them to sequentially increasing threatening stimuli, with Appendix A being the least threatening, and Appendix A being the most threatening [30]. Each of the first three videos was looped twice and ran for 25 s in total, while Appendix A was not looped and ran for a similar time. Participants were shown the series of four videos, from least to most threatening. This was to ensure participant well-being, guided by pilot group testing and feedback, which had shown a potential for a significant negative reaction to the later injury videos. By keeping the videos sequential, we ensured that any adverse reactions were identified in the earlier videos and minimised.

Appendix A ‘Running’: shows the video athletes running in a straight line at a low to moderate intensity for 10 s, followed by a gentle turn to leave the video frame -this was considered a baseline video showing a common, very lower-level activityAppendix A ‘Cut and Pivot’: shows the video athletes running forward at a low to moderate intensity for nine seconds, then completing a cut and pivot movement (i.e., planting one foot and turning sharply at a right angle) then continue off screen—this is the most commonly reported fearful movement for people following ACL injury [30].Appendix A ‘Cut & Pivot with feigned knee injury’: shows the video athletes running forward at a low to moderate intensity for nine seconds, then completing a sharp cut and pivot motion (as in Video 2) and then feigning a knee injury by holding the knee and falling down (reproducing an injury).Appendix A ‘Knee injury clips’: shows a series of video clips of traumatic knee injuries compiled from YouTube, representing a series of more graphic injuries (both male and female)

YouTube links to videos can be found in Appendix A.

A guiding principle in the creation of the videos was to avoid bias that might affect participants’ responses. Potential biases affecting an individual’s fear/distress included (but were not limited to) the setting/location, the backdrop of the movements, and participant specific factors (e.g., age, gender, body type, and clothing). To minimise bias, filming of Appendix A was completed in a grassy location with an unremarkable backdrop of trees in the medium-far distance. This location was selected to not emulate a sports field, as a specific field may have resonated more with one participant over another. One male actor and one female actor, aged around 30 years, of average build, were selected for the videos. Both wore the same plain, unlabelled, unisex, neutral (blue and black) clothing (Figure 1). Videos were edited to show both the male and female simultaneously in a split-screen format. This was to allow the participants to watch/identify with either male or female actor.

### 2.6. Qualitative Data Analysis

All interviews were performed by CL, a titled physiotherapist with experience in patient centred interviewing and PhD candidate, who has experience in elite sport and ACL injuries both personally and professionally. Interviews were audio-recorded, transcribed verbatim, entered into NVIVO 12 (QSR International, Melbourne, Australia) and analysed using an interpretive descriptive framework [51]. Qualitative data from each video were pooled for analysis to allow for an overall presentation of the fear response. The methodological steps were closely followed throughout the analysis [52]. This involved two coders CL and CS, a titled physiotherapist and PhD candidate with experience in the clinical management and research of ACL injuries, independently coding the data. The coders used an inductive open coding method whereby codes were identified from the data and not preconceived. Inductive coding was first completed on 12 transcripts. Coders then discussed and synthesized their codes to create a single codebook and retrospectively applied this codebook to the first 12 transcripts, paying particular attention to any concepts that were not captured by the codebook. Once the coders were satisfied that the codebook accurately captured all relevant raw data, the codes were then presented to the broader research team for further discussion: DB, a specialist musculoskeletal physiotherapist and senior research fellow with expertise in the biopsychosocial nature of musculoskeletal disorders, CM, a Sport and Exercise Psychologist with over 20 years expertise in stress and anxiety in sport and exercise, and AL, a Senior researcher and lecturer in sports science. Through group discussion and negotiation, codes were further refined. Coders then completed coding of the remaining interviews to test, challenge, and refine the final themes.

### 2.7. Quantitative Data Analysis

Quantitative analyses were performed using Stata 16.1 [53]. Fear and distress were treated as separate dependent variables and modelled separately. Data was checked for normality with the Shapiro–Wilk test [54], with the majority of variables reaching the threshold for normality. Linear mixed models with random subject effects were used to analyse the relationship between both fear and distress for the four video conditions. Margins plots were created and post hoc pairwise comparisons (Tukey’s test) were used to compare the differences between the individual videos. Results were reported as the estimated mean difference, 95% Confidence Intervals (95% CI) and the associated *p* value.

## 3. Results

### 3.1. Participant Characteristics

A total of seventeen participants 1-year post-ACL injury/surgery with high fear on the TSK were included (Figure 2, Table 1). The average age was 28 years, 71% were female, with an average time since their last ACL injury of 5.5 years. Sixteen of the seventeen participants had their injury managed surgically. Ten out of the seventeen participants returned to competitive sport at some level. Further demographics about the participants injury and sport is listed in Appendix A.

### 3.2. Qualitative Results

Five themes were identified from the qualitative responses to the threatening stimuli (see Table 2 for supporting quotes—these are referenced in the relevant text, full transcripts are available in Appendix A).

#### 3.2.1. Theme 1: Evoked Physiological Responses

Some participants reported physiological reactions (i.e., automatic, instinctive, unlearned responses) when watching the feigned injury (Appendix A) and the knee injury videos (Appendix A). These participants described a range of involuntary reactions, including a muscle ‘twitch’ (Quote 5, Q5) or a ‘tensing’ around the knee (Q2), as well as feeling ‘sick’(Q1). A few participants also reported feeling pain in the knee while viewing the videos, attributing this pain to psychological drivers such as ‘anxiety’ (Q3) or a ‘phantom pain’ (Q4). Participants reported wanting to physically reassure themselves (Q4) by ‘touching’ their knees (Q3–Q5).

#### 3.2.2. Theme 2: Deeper Contextualisation of the meaning of an ACL Injury Influencing Body Confidence

Participants discussed beliefs about their own ACL injury in response to watching the threatening videos. They spoke of unhelpful extrinsic messages from those around them, such as those given by medical professionals, friends and/or societal/media messages, which shaped their evaluation of their knee stability and susceptibility to injury (Q6). These participants reported experiencing high levels of fear, which they attributed to beliefs of weakness and vulnerability of their own ACL-injured knee. For example, Participant 8 stated that any return to sport (after an ACL injury) involves risk of re-injury and there is an expectation that something will ‘go wrong’ (Q8), a message he gleaned from his rehabilitation team. Intrinsic messages of the perception of their knee and beliefs about knee function were shaped by personal experiences. Beliefs about weakness of the knee, and susceptibility of the knee to a re-injury (Q7), resulted in reduced confidence.

#### 3.2.3. Theme 3: Recall of Psychological Difficulties

Most participants reported psychological effects experienced during the injury journey when viewing the videos. For some participants, the video exposure resulted in a personal recount of their own injury such as, bringing back ‘bad memories’ (Q9, Q13), or remembering the moment of their own ACL injury, the ‘exact feeling of it’ and the ‘noise’ (Q14). Others discussed how ‘confronting’ (Q10) and fearful the movements in the video (referring to the plant and cut movement) were to them in the present time (Q11, Q12).

#### 3.2.4. Theme 4: Negative Implications of a Re-Injury

When watching the feigned injury and the series of traumatic knee injury videos, participants thought about what effects a re-injury would have on their own quality of life. Participants expanded on what the ACL injury had meant to them, not just from the pain and injury but from the wider psycho-social consequences (Q16, Q17). The negative consequences of a re-injury and factors that influenced fear appraisal included: rehabilitation (Q17, Q18), socio-economic implications (Q17), and changes to an individual’s function, life, and identity (Q20) ‘for the next few years’ (Q19).

In response to the feigned injury and the traumatic knee injury video, some participants reported how their previous ACL injury had changed their sporting lives (Q21–Q23). They reported a loss of athletic identity because they felt they were no longer capable of playing at their previous level. This had a negative impact on their sporting activities and overall physical confidence. For example, one participant indicated a perceptual shift from being the person who would ‘say yes to anything’ to becoming avoidant of physically challenging activities (Q22). These participants expressed increased fear and negative emotions related to not being able to play sport again; ‘yeah, I was crying in my bed some nights’ (Q21).

### 3.3. Quantitative Results

#### 3.3.1. Self-Reported Fear

Figure 3 depicts the individual and group level changes for the rating of fear with exposure to the four levels of increasing threat. The quantitative results indicated a significant increase in the fear rating for each increment in the four videos. The largest increment between exposures for fear was found when participants moved from observing the cut and pivot (Appendix A) to the feigned injury (Appendix A) with an estimated mean difference of 3.1 on a scale of 0–10 (95% CI 2.1–4.0, *p* < 0.001). (Linear Mixed Models Analysis is provided in Appendix A). 

#### 3.3.2. Self-Reported Distress

Figure 4 depicts the individual and group level changes for the rating of distress with exposure to the four levels of increasing threat. The linear mixed model results indicate a significant increase in distress for each increment. The largest increment between exposures for distress was also observed from cut and pivot (Appendix A) to the feigned injury (Appendix A) videos with an estimated mean difference of 3.2 (95% CI 2.3–4.2, *p* < 0.001). (Linear Mixed Models Analysis output is provided in Appendix A).

## 4. Discussion

The purpose of the current study was to improve understanding of fear and distress, using qualitative and quantitative methodologies, within an ACL-injured cohort with above average fear during exposure to videos depicting sequentially increasing threat to knee stability. In doing so a range of responses (qualitative themes) were identified. The themes including physiological reactions to viewing the videos (such as participants rubbing their knee to soothe and protect themselves), contextual information affecting confidence, negative psychological associations, concerns about the ramifications of a re-injury and changes to their athletic identity. These themes were complementary [32] to quantitative ratings of fear and distress, in that with both the qualitative and quantitative outcomes, witnessing an injury (feigned or real) evoked the strongest responses, with the qualitative themes adding depth and breadth to the quantitative data.

### 4.1. Comparisons to the Literature

A number of studies using qualitative methodology to investigate the lived experience of an ACL-injury have touched on the concept of fear of re-injury [55,56,57,58,59]. These studies largely explored the practical consequences of the injury such as fear of surgery, the rehabilitation journey and socioeconomic considerations of a re-injury [56,57,58,59,60]. Unlike those prior studies, this study used videos to elicit a real-time response to a perceived threat, as opposed to asking people with an ACL-injured knee to recall their experience. Despite this difference in methodology similar constructs emerged in our data (Table 2) including fear of the rehabilitation process [56,58,59], social changes [56,57,58], and changing identity [57]. Similarities between the qualitative themes of this study and other non-qualitative research can also be identified. A range of factors such as fiscal position and time off work [61], the level of trauma associated with the initial injury [10], the duration and difficulty of the ACL recovery process [39,62] and rehabilitation [63] have been previously identified as factors which can affect return to previous level of function.

During testing, participants reported a range of psychological effects from viewing the videos including anxiety and rumination (Table 2). The substantial psychological effect of an ACL injury has also been identified in other research [61,63,64,65]. The combined importance of psychological, social factors and physiological changes seen in this cohort supports the notion in this broader literature that an ACL injury results in a wide range of effects which can drive fear and distress.

The quantitative results for fear and distress were broadly equivalent (Appendix A, Figure 3 and Figure 4), suggesting it may be difficult to untangle fear from distress within an ACL injured cohort. Despite psychological effects being universally reported, only 2 out of 17 participants were asked by their healthcare professionals about their mental health during the rehabilitation process (Table 1). Those enquiries were in relation to return to sport performance, not related to the broader emotional and psychosocial health of the participants. Positive psychological support has been shown to be important in managing fear following an ACL-injury [66]. As fear of re-injury is a significant barrier for return to sport after an ACL injury [7,8,14,15,16,17], our findings support the notion that healthcare professionals can overly focus on the physical aspects of surgical repair and rehabilitation stages, while the issue of psychological support is often not considered.

There were indications of messaging from society and healthcare professionals that contributed to the beliefs underlying fear (Table 2). This is consistent with observations related to other musculoskeletal conditions such as persistent hip pain [67] and chronic low back pain [68]. Messages of damage, weakness and vulnerability, and messages reducing an individual’s control of their injury can all have a significant negative effect on functional outcomes [67,68]. There is a greater need to understand how messages from society, close contacts and healthcare professionals might contribute to fear after an ACL injury. Further, these messages seem to have a link to the individual’s athletic identity. Some individuals with stronger athletic identity report more emotional trauma following an ACL injury due to the increased challenge to their self-purpose [10,65], although this association is complex [69].

Perceived physiological reactions reported in this cohort may have been driven through activation of the sympathetic nervous system (SNS) in response to the perceived threats [70]. Similar responses (in the SNS) have been observed in people with other conditions and injuries when exposed to threatening stimuli, such as anxiety disorders [71] and chronic low back pain [72]. Within this cohort, increased heart rate, sweating, faster breathing and feeling ‘distressed/sick’ (e.g., Quote 1) could all be signs of a SNS reaction [20]. It should also be noted that these physiological responses could represent an empathic reaction of the participant towards the actors when viewing the videos [12]. Functional Magnetic Resonance Imaging studies have indicated that similar areas of the brain are involved during pain and pain empathy, and the latter can be activated when watching a painful scenario [73]. Further research comparing people with high and low fear after an ACL-injury exposed to a threatening video stimulus may provide further insight into the specific contribution of ‘pain empathy’ versus the heightening of fear from an ACL injury. Assessing biomarkers in the brain [74] may also assist in understanding the physiological sequelae of watching the threatening stimuli.

Fear in this cohort is likely to be complex and multifactorial. The participants often exhibited defensive, protective mechanisms in response to the videos, a behaviour that may be explainable through the concept of fear conditioning [75]. Responses within this cohort supporting fear conditioning [25] included behavioural (e.g., avoidance of an activity), perceived physiological (e.g., increased muscle tension) and cognitive responses (e.g., catastrophizing the injury or recalling the psychological difficulties). The Fear Avoidance Model (FAM) [76] may help to explain this cohort’s behavioural cycle. The FAM has been used to explain the cycle of fear, avoidance and disuse in conditions such as chronic low back pain [23]. In the FAM, a range of biological, psychological, and social factors following an injury may heighten pain and dysfunction, further reinforcing fear (and avoidance), thus creating a cycle of chronic disability and suffering [23,24]. Although pain, a driver in the FAM, was not a commonly reported issue within this cohort, a range of protective biopsychosocial factors, which heightened fear and avoidance could be seen [77,78].

### 4.2. Strengths

The use of videos represents a novel, low-risk method to assess fear of re-injury and distress in people who have experienced an ACL-injury. This was performed in real time, rather than relying on patient recall, with self-reported injury data being generally poor [79]. This may allow clinicians and patients to enter into a dialogue around a patient’s emotions and develop strategies if they are a barrier to function.

### 4.3. Limitations

Participants were included in this study via a designation of ‘above average fear’ dependent on their score on the TSK [17]. However, this questionnaire, despite being commonly utilised within the ACL literature, does not appear to have been fully validated in people following an ACL-injury. To the best of our knowledge there is no current psychometric scale that specifically assesses fear following ACL injuries. The multi-faceted nature of fear may not be adequately measured using a psychometric scale that is focused only on fear of painful movements (TSK), as evidenced by the broader variety of fear responses seen in this study. Within the development of the videos, we attempted to reduce any potential bias that would affect results. However, we acknowledge the likelihood of inherent biases which may impact different people (e.g., age, ethnicity and mechanism of injuries). Future studies with a wider diversity of participants would better help the generalisability of these videos for a broader cross-section of society.

The order of the videos may have been a limitation in the study. We considered the possibility of an ‘order effect’ of increased participant scores to the later videos due to the sequential nature of the videos. However, through consumer engagement we determined that participants may have strong adverse effects to watching the fearful stimuli. Keeping it ordered from least to most threatening allowed participants to exit the testing procedure at any point of time if required, improving participant safety. Future studies may use video randomisation to reduce order effects.

Our final sample included one person managed conservatively. While it was not the aim of this study to compare people managed surgically versus conservatively, given similar long-term outcomes for both groups [37,38,39,40,41,42,43] it is possible that the responses in these two groups might differ. We do not know if the responses provoked from the videos are specific to people with ACL injuries or are results that could be mirrored in the general population or in people following other lower limb injuries. Further work is needed to investigate the responses to the video stimuli in these alternate cohorts to determine the relevance of the results to an ACL injured cohort.

### 4.4. Clinical Significance

This study provides a novel way of assessing fear in people who have experienced an ACL injury. The video testing procedure is relatively quick to administer and provides a useful measure that could potentially be used for re-assessment purposes. This video procedure may be particularly useful in later stage rehabilitation. Showing a video in a clinical environment may provide additional insight compared to speaking about it in isolation. Further, if questioned in a clinical setting the person concerned may be guarded in their responses or not sufficiently self-aware of their psychological barriers to be able to answer effectively. The results of this study indicated that exposure to the videos initiated and promoted an open discussion, particularly in relation to the psychological barriers effecting return to activities. Clinically, this may lead to improved identification of these barriers and refinement of rehabilitation programs.

Of note was the identification of a number of subjects who had physiological reactions to the videos. To the best of our knowledge this has not been reported in other ACL and fear literature. Physiological responses may be a strong indicator of subconscious fear conditioning, the identification of which may further assist in fashioning effective clinical management strategies.

## 5. Conclusions

Fear is a significant barrier to optimal outcomes after an ACL injury. This study, using qualitative and quantitative methodology, has shown that people with an ACL injured knee can report significant fear and distress when shown videos depicting threat to their knee. This cohort reported a broad range of themes including (1) Evoked physiological responses, (2) Deeper contextualisation of the meaning of an ACL injury influencing bodily confidence, (3) Recall of psychological difficulties, (4) Negative implications of a re-injury, and (5) Change to athletic identity. Viewing a feigned or real injury appeared to evoke stronger responses than activities performed without an injury occurring. The fear narrative is participant-specific. The results support the notion that in addition to being a physical injury, an ACL injury has more complex neurophysiological, psychological, and social characteristics which should be considered in management. Using video exposure in the clinic may assist identification of underlying psychological barriers to recovery following an ACL injury, facilitating person-centred care.

## Figures and Tables

**Figure 1 sports-10-00183-f001:**
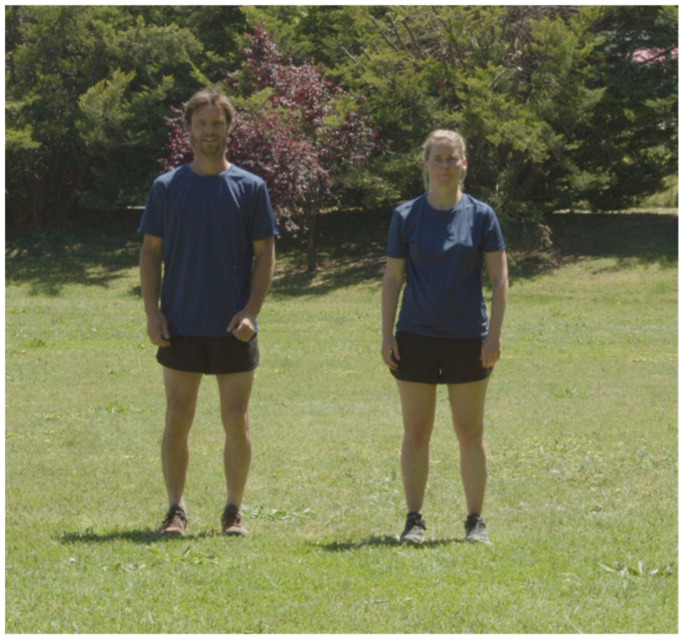
Video athletes.

**Figure 2 sports-10-00183-f002:**
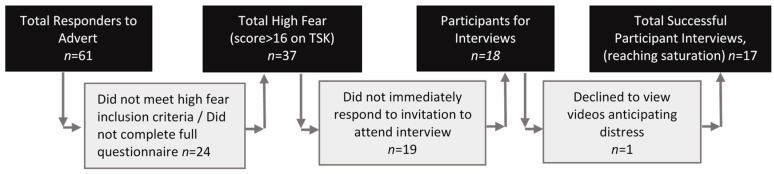
Flow diagram for participant recruitment.

**Figure 3 sports-10-00183-f003:**
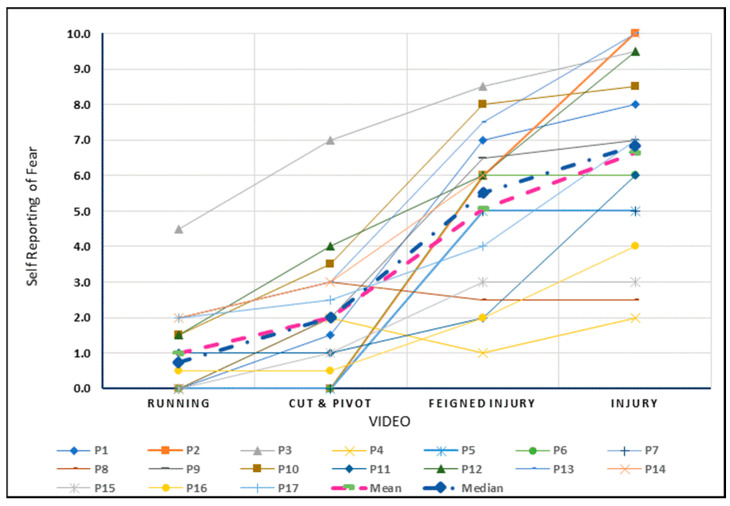
Self-reported fear viewing videos depicting increasing threat to the knee.

**Figure 4 sports-10-00183-f004:**
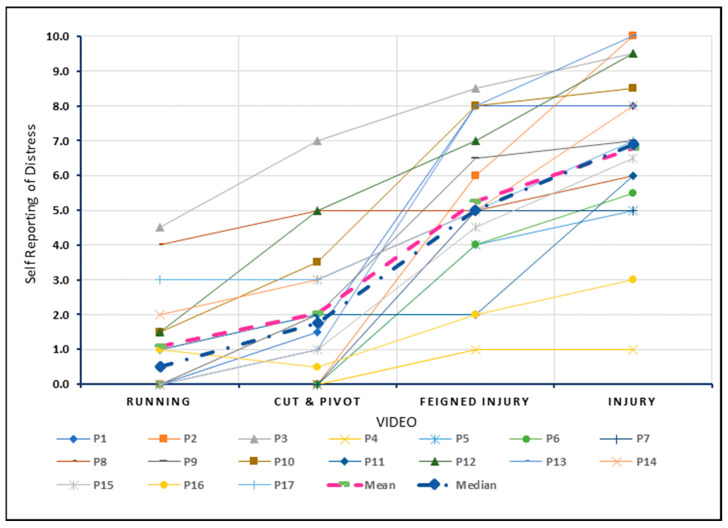
Self-reported distress viewing videos depicting increasing threat to the knee.

**Table 1 sports-10-00183-t001:** Participant Data.

ID	P1	P2	P3	P4	P5	P6	P7	P8	P9	P10	P11	P12	P13	P14	P15	P16	P17	Mean (SD)	Median (IQR)
Age	22	25	33	28	24	22	28	32	23	31	33	32	31	24	23	36	25	27.70 (4.58)	28 (23.5, 32)
Sex	F	F	M	F	F	F	M	M	F	F	F	F	F	M	M	F	F	M:5, F:12	
Tampa Scale of Kinesiophobia ^1^	19	17	33	23	23	18	21	17	17	31	25	17	22	25	30	18	30	22.71 (5.5)	22 (17.5, 27.5)
ACL-Return to Sport Index (ACL-RSI) ^2^	39.4	34.2	25.8	41.7	52.5	28.3	53.3	63.3	66.7	20	16.7	65.8	18.3	31.7	47.5	41.7	15.8	38.98 (17.21)	39.4 (22.9, 52.9)
Fear of movement ^3^	1	4	3	0	4	2	0	0	0	10	0	0	6	3	8	2	2	2.65 (3)	2 (0, 4)
Distress/Anxiety related to knee movements ^3^	1	1	5	1	4	0	0	0	0	8	5	0	6	4	7	3	1	2.71 (2.73)	1 (0, 5)
Lack of Confidence in Knee movements ^3^	0	8	7	2	5	2	0	0	0	10	5	0	6	4	6	5	4	3.76 (3.15)	4 (0, 6)
Pain ^3^	1	7	2	1	6	0	2	0	0	3	0	0	6	1	2	3	1	2.06 (2.28)	1 (0, 3)
Surgical reconstruction	Y	Y	Y	Y	Y	Y	Y	Y	Y	N	Y	Y	Y	Y	Y	Y	Y	Y:16, N:1	
HCP Provided Psychological Support ^4^	N	Y	N	N	N	N	N	Y	N	N	N	N	N	N	N	N	N	Y:2, N:15	
Return to Previous level of function ^5^	2	2	0	3	3	2	3	3	3	2	1	1	1	2	1	1	1	0: 1, 1: 6, 2: 5, 3: 5	
Year of ACL injury	‘16	‘17‘18	‘16	‘16	’17	‘11‘16	‘17	‘11‘12	‘15	‘15	‘08	‘08	‘06‘09‘13	‘19	‘19	‘06	‘19		

^1^ Tampa Scale of Kinesiophobia. This scale was rated 0–51, with a higher score indicating higher levels of fear. ^2^ Anterior Cruciate Ligament Return to Sport Index. This scale rated out of 100 with lower scores indicating higher impairment. ^3^ “In relation to your knee, in the last week how much have you been bothered by it?” (rated 0–10, 10 indicating extremely bothered). ^4^ HCP: Health Care Practitioner. ^5^ Return to previous level of function scoring: 0–3 where 0 = no return to any exercise, 1 = Return to basic exercise (gym, etc.) but no or minimal sport (i.e., non-competitive), 2 = Return to sport at a lower competition level, 3 = Return to previous level of sport.

**Table 2 sports-10-00183-t002:** Supporting quotes for Thematic Analysis.

**Theme 1: Evoked Physiological Responses**	
Quote 1 (Q1)	“I honestly stopped watching after the first rebound, it made me feel sick …”. (Participant 2; P2)
Q2	“Like my body tenses and I sort of imagine that that’s your knee, and then your muscles will kind of tense up unconsciously to protect it. (P6)”
Q3	“I get anxious when I see someone go down screaming in agonizing pain holding their knee knowing that it may be an ACL. I feel that pain but that’s only in terms of that stress and anxiety… I do grab at it (my knee) if I see someone lunging and grabbing at their knee. It’s weird.” (P14)
Q4	“The immediate reaction that I have sometimes when you see other people tear their ACL is you want to go, “Oh, shit.” You physically check, “Is my knee still intact?” [laughs]. … It’s a phantom pain that you remember because it’s not that I’ve experienced that many super painful things in my life.” (P15)
Q5	“Every time I see someone go down with a knee (injury), it doesn’t even have to be an ACL but I feel my knee twitch. If I see someone go down holding their knee, I will sometimes hold my knee, as well.” (P15)
**Theme 2: Deeper Contextualisation of the Meaning of an ACL Injury Influencing Body Confidence**	
Q6	I think with the distress or how worried I am or was when I did it, a lot of that was already built up. If that makes sense. If I didn’t know nothing about it, or hadn’t seen it happen (referring to messages and clips in the media around snow sports), or had people that I knew do it, it probably wouldn’t have been as bad for me. I think about that a fair bit then I just worry about it more than I should. Does that make sense? (P3)
Q7	I guess that (cut and pivot movement) made me feel like he should be careful doing that because that’s how you can get ACL injuries, changing directions really fast. (P12)
Q8	If you want to play sports and push it hard you have got to expect that at some point something is going to go wrong…there is a fear that you are going to hurt yourself I think everyone has that. (P8)
**Theme 3: Recall of Psychological Difficulties**	
Q9	It (series of traumatic knee injuries video) gives you bad memories and makes you not want to play sport again I reckon. Watching a compilation of ACL injuries especially the way that you did your one makes you think maybe you shouldn’t go back to playing netball again because that’s what’s going to happen, or that’s what could happen. (P12)
Q10	It’s quite confronting because that’s probably what I looked like (when getting injured). (P13)
Q11	I feel like I’m cautious watching them. I’m fearful of thinking of myself doing that (plant and cut action). (P17)
Q12	I used to watch people change direction, it would give me that like, ‘uh, can’t imagine doing that’. (P9)
Q13	“Ooh”. It was distressing to watch them because it looks so painful (referring to the cut and pivot + feigned injury and a series of traumatic knee injuries). A couple of times, while playing, I’ve ended up in something similar, so it’s more like a little bit anxiety as well, just feeling anxious about it because it brings back—I don’t know, bad memories. I distinctively remember falling on the ground and just touching my knee in pain in a similar situation. (P10)
Q14	I feel like I just remember the exact feeling of it. The classic empathy kind of thing. I remember the exact feeling. How terrible it was. When I see those videos, I actually think more of the noise, (referring to the noise when sustaining ACL injury), I distinctly remember the noise. When I see those videos, I almost see the noise. (P17)
Q15	I find that they (thoughts of injuries) just pop into my brain randomly and it makes me really uncomfortable because you’ve got to try and get it out (from your head) and think about something else which is why I really hate watching them because I don’t need … I just don’t want new fresh ones (thoughts) that decide they want to come in every now and again. (P5)
**Theme 4: Negative Implications of a Re-Injury**	
Q16	I just feel bad for them knowing what they now have to go through (after watching cut and pivot with feigned injury). (P7)
Q17	Already being on edge as I am, doing that and then just knowing that, “All right, well, should I try and give up competitive sports?” That’s the way I think, I just don’t want to do it. From the pain associated with it, the rehab and the time it takes and, like I’ve touched on, the financial implications of it as well. I always sympathize with the athletes when I see someone go down … you just see that (traumatic injury video), and you say, “Well there goes the next year of your career.” It’s the trauma, the injury, but also how it affects you, how it affects your livelihood, and whatnot. (P14)
Q18	I guess, elite athletes when they do an injury, I know it’s their entire life, but that, they do have an entire team of people straight away onto it, doing everything for them to help them out. When its someone, a poor old pleb, they don’t realize how much of an impact it might have. (P4)
Q19	You don’t want to see anyone’s knee bend that weird … that second of them thinking, “Oh, shit, my life is going to change considerably after this one second for the next few years. My life choices have changed, and my options are limited.” (P15)
Q20	Feeling my joint absolutely just give way. It’s pretty messy, it’s not fun. You’re like, “That person’s really good,” and then being injured, and it’s like, “Oh, shit. That person’s life is going to change forever now.” It’s a pretty big deal. (P15)
**Theme 5: Change to Athletic Identity**	
Q21	I do remember some nights just crying in bed, like not too many … because as I said before playing sport every day doing it because I love it so much and not being able to do it. Yeah, I definitely was crying in my bed some nights. (P1)
Q22	I used to be the person that would say yes to anything. Anything sports-related, I would just say, “Yes, let’s do it, give it a try”. Now I don’t. (P11)
Q23	But, on the court, I was mostly slower and less agile. I think that was just fear…I’ve convinced myself that I never will play as well, and that’s why I’m not doing it. (P2)

## Data Availability

The data presented in this study are available on request from the corresponding author. The data are not publicly available due to privacy reasons.

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
