# Peer review of "An Investigation of the Nature of Fear within ACL-Injured Subjects When Exposed to Provocative Videos: A Concurrent Qualitative and Quantitative Study"

_sports, 2022, doi:10.3390/sports10110183_

Round 1

Reviewer 1 Report

The fact that the study was submitted to an ethics committee and that it was approved does give peace of mind.

The proposed manuscript, is very interesting, and very well thought out. Congratulations. My comments follow below.

Comment 1

The link for Video 4 is missing.

Comment 2

Why was Significance set ap p<0,01?

Comment 3

Are there other collected  data from the participants of this study that can add some more information specifically to previous level of sport activity, past injuries, and expected RTP considerations?

Comment 4

Please make it clear how “Videos depicting increasing threat to knee stability might be useful in assessing fear during the return to sport phase stage of rehabilitation following an ACL injury”.

Reviewer 2 Report

In the abstract and in the methods:

It s necessary to define the eligibility of the sample.. Have you quantified this "average" fear? If not, then do you omit? Also because in the methods part of the abstract, there should not be the result of the inclusion, but the description of the population to be reached ...

Did they have a reconstruction? Do they practice sports? If so, amateur? Pro?

I would put in the title the concept of qualitative analysis ...

Line numbers are missing ... however when you say "Quantitatively, participants reported consecutively greater levels of fear and distress with the increasing threat", better to say that there is direct proportionality

Conclusions, why? To exorcise fear? Provide a reason that ties your suggestion to the resultsIn the introduction I would add that “The incidence of anterior cruciate ligament (ACL) tear is 2.8-3.5 times greater in women than in the man and the ACL injuries commonly occur in noncontact situations during direction changes, cutting maneuvers, or during landing after jump” reference: http://doi.org/10.7752/jpes.2020.05342

“A mixed methods design was selected to allow for both qualitative and quantitative”, if this is the approach then I would reformulate the abstract

Figure 1 is a result.

Table 1 is a result .. in addition, the scales used must be described in the methods after the design and the intervention .. before this type of table make an overall one with means and SDs. Dividing between who has had an ACL-R and who has not.

I recommend the sequence in the methods: Design, Population, Intervention and Outcome. Remove any results from the methods

Self-reported Fear, is the almost primary outcome .. describe it in the methods on how it was assessed ..

When you finish the discussion, by convention you just put the limitations standalone.

Refer exclusively to the results obtained in the conclusions, the fact that the subjects are afraid as well as taken for granted is a criterion for inclusion! You have not deepened the concept between those who have reconstructed the ligament or not, you have not recalled the themes analyzed.

Round 2

Reviewer 2 Report

The abstract has improved, but in the conclusions, I would suggest a clinical impact, otherwise the work is an end in itself.

96 Is the focus on the nature of fear? Add it in the title

204-219. The link to youtube is fine but for an unconditional usability by third parties, I would put the videos as supplementary material

Very detailed results and improved figures thank you.

Author Response

Response to reviewer 2

Response before 07/11/2022

Reviewer comment

Reply to the reviewer comment

Changes made to the text

The abstract has improved, but in the conclusions, I would suggest a clinical impact, otherwise the work is an end in itself.

Thankyou, for the comment, we have changed the end of the conclusion as requested.

Previous: Videos depicting increasing threat to knee stability might be useful in assessing fear during the return to sport phase stage of rehabilitation following an ACL injury. Line 35-36

Current: The results support the notion that in addition to being a physical injury, an ACL injury has more complex neurophysiological, psychological, and social characteristics which should be considered in management. Using video exposure in the clinic may assist identification of underlying psychological barriers to recovery following an ACL injury, facilitating person-centred care. Line 35-38

96 Is the focus on the nature of fear? Add it in the title

Thankyou.

We have added it into the title as requested

Current: An investigation of the nature of fear within ACL-injured subjects, when exposed to provocative videos: A concurrent qualitative and quantitative study. Line 2-3

204-219. The link to youtube is fine but for an unconditional usability by third parties, I would put the videos as supplementary material

Thankyou,

We have moved the links to the videos as requested.

Current: YouTube links to videos can be found in supplementary material.

Very detailed results and improved figures thank you.

Thankyou
